# Evaluation of Vision Language Models with Item Response Theory

## Abstract

Evaluation of generative AI output is difficult because of the high dimensional
nature of the problem space. Accuracy-oriented benchmarks are often used to
assess the quality of outputs, but may not provide a complete picture because they
do not incorporate the difficulty of items relating to a particular task. We present
the use of Item Response Theory (IRT) to evaluate the output of a cohort of Vision
Language Models (VLMs) on two different tasks: image caption rating, and visual
reading comprehension. As a result, we show how to find meaningful differences
between popular state of the art VLMs, promoting meaningful interpretability.
IRT can be used in many areas of the ML workflow. We will cover some of the
prior work and ways IRT may be used in your research. Our aim is to encourage
the adoption of IRT by the ML community as a general tool for evaluation.

## 1 Introduction

In this work, we showcase the use of IRT for evaluating latent abilities of ML models. We will first
review the basics of IRT and Wright Map Analysis, and highlight some previous work related to
ML. We will then apply IRT to two case studies involving evaluation of VLMs on different tasks:
caption rating for visually impaired and blind people Scott et al. (2023); Narins et al. (2023), and
visually-informed comic reading comprehension Blum et al. (2020). We will then suggest other
ways that IRT may be used in ML workflows and generative AI research. Our hope is that these
techniques will be adopted and broadened by the ML community to aid interpretability, validation,
trust, fairness, alignment, and bias of machine learning models, datasets, and scales.

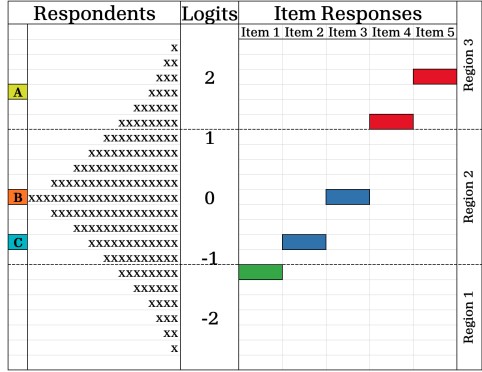

Figure 1: General form of Wright Map, here shown with 5 items and several respondents. The
area on the left is the distribution of the respondent's ability, and the area on the right is the item's
difficulties. They are both plotted on the same logit scale, providing probabilistic interpretation.

Traditional ML metrics for LLM and VLM development focus on optimizing the performance of
the model, and assumes that each sample in the dataset is equally difficult to infer. When we want to
find out which model is best at a natural language processing (NLP) task such as question answering
(QA), visual question answering (VQA), captioning, image-caption rating, semantic similarity, or
sentiment analysis, we often use classification metrics or NLP metrics. Metrics such as accuracy,

precision, recall, F1-score, BLEU Papineni et al. (2002), METEOR Denkowski & Lavie (2014), ROUGE Lin (2004), and CIDER Vedantam et al. (2015) are used to cross-compare the performance of top models on common datasets, but they do not consider the differences in individual samples; IRT does account for those individual differences when assessing the model performance.

The main contribution of this paper is a new methodology and benchmark for evaluating VLMs using IRT and Wright Map Analysis Wright (1982). The methodology enables us to determine how well VLM respondents perform relative to humans, and the benchmark establishes a quantitative and interpretable process for evaluating progress in VLM research. We cover two specific case studies and analyze them from an IRT lens. The results of the paper provide useful information to researchers working on evaluating model outputs, and for generating synthetic data for augmenting datasets. The results may also be useful for researchers who are collecting data from human raters and want to determine validity of using VLMs as supplemental raters, as a possibly faster, alternative to human-annotated large datasets.

## 2 ITEM RESPONSE THEORY

Item Response Theory is a methodology used in Psychometrics to study psychological traits and abilities Lord & Novick (2008). It uses the terms *items* and *respondents* to refer the individual samples and persons. As the name implies, IRT takes into account the characteristic behavior of the item when assessing performance. Specifically, IRT measures respondent's *ability* in the presence of or with regard to an item's *difficulty*, providing a more nuanced view of performance. IRT is often used in standardized testing, such as the GMAT and GRE tests Murodulloyevich (2025), and is desired when optimal decisions are necessary, and interpretability is essential.

For example, imagine giving a test of 100 multiple-choice problems to 100 students. Each student has their own accuracy measured by the percentage of problems they got correct, but the problems also have an accuracy measured by the percentage of students that got the problem correct. A problem's accuracy reveals something about its difficulty: problems that everyone got correct are too easy, and a problems everyone got wrong are too hard. It is possible that, when grouped by difficulty, there may be an imbalance in the percentage of easy problems that were correct versus the percentage of hard problems that were correct. This could lead to a situation shown in Table 1.

If we select the best model based on Total Accuracy, then we would select Student A, but that would also mean we would inadvertently be optimizing for Easy problems.

| Student | Total Acc. | Easy Acc. | Hard Acc. |
|---------|------------|-----------|-----------|
| A | 93% | 100% | 86% |
| B | 92% | 91% | 93% |

Table 1: Accuracy breakdown by item difficulty. Which student did better?

IRT considers the item difficulty along with the person's ability and places them on the same probabilistic, interval scale (i.e. log-odds, or logits). Placing both items and persons on the same interval scale allows for a more detailed level of analysis because the magnitudes of the differences between items and persons have probabilistic meaning, and may be considered in context of the latent trait being modeled. To do this, we apply a one parameter logistic model (1PL), also known as a *Rasch Model*. The 1PL equation is given by:

$$P(X_i = 1 | \theta, \delta_i) = \frac{1}{1 + e^{-(\theta - \delta_i)}} \qquad (1)$$

where $\theta$ is the person's *ability*, and $\delta_i$ is the *difficulty* of item $i$ Rasch (1960); Wilson (2023). The values for $\theta$ and $\delta$ are found by using statistical inference methods such as Maximum Likelihood Estimation (MLE), Monte-Carlo Estimation, or Variational Inference, which also generates item "fit" statistics that indicate how reliable the estimates are Wu & Adams (2013). In this work, we used the partial-credit model (PCM), which is an an extension to the 1PL that preserves the levels in polytomous data. That equation is given by:

$$P(X_i = k \mid \theta, \boldsymbol{\delta}_i) = \frac{\exp\left(\sum_{j=0}^{k}(\theta - \delta_{ij})\right)}{\sum_{h=0}^{m} \exp\left(\sum_{j=0}^{h}(\theta - \delta_{ij})\right)} \tag{2}$$

where $\delta_{ij}$ is the difficulty of item $i$ at step level $j$ Wright (1982); Wilson (2023). We focus specifically on the 1PL, rather than 2PL and 3PL, because of its interpretative power of placing respondents and items on the same logit scale, and the fact that the WrightMap is only defined for 1PL.

Since $\theta$ and $\delta$ are calibrated on the same logit scale, they can be plotted relative to one another on an *item-response map*, also known as a *Wright Map* Wilson (2017), which is a compact and very informative visualization for communicating Rasch-family model output. Mark Wilson coined the term "Wright Map" in honor of Ben Wright who was a proponent of their use and expressive ability, and had extended them in many ways Wilson (2011). We use them for their expressive interpretability, identifying meaning in the latent space we are studying.

The Mean Square fit statistics Wu & Adams (2013) is a quantitative way to evaluate the validity of the Wright Map. The ideal range for the fit statistics is between 0.77 and 1.33, where 1.0 is the ideal score. If an item has a fit statistic greater than 1.33, it would suggest too much randomness in the item; that is, people with high ability are getting it wrong, and people with low ability are getting it right. A fit statistic below 0.77 suggests overfit, that the item is too predictable. These fit statistics are checked to be within acceptable range before accepting the validity of the Wright Map.

Figure 1 shows a generic Wright Map. The key features of the Wright Map are the person abilities, plotted on the left as a histogram or distribution, the mean item difficulties, plotted on the right at their individual locations, and the interval-level logit scale marked in the middle and showing the y-axis of the graph. The dotted lines are *waypoints* that signify theoretical or substantive regions of interest. They are a visual indicator showing certain items correspond to certain regions. The letters "A", "B" and "C" on the far left show particular respondents in which we are interested in studying. The "x" marks represent the entire sample studied, including respondents of interest.

Once the Wright Map is generated, interpreting the result is like reading a map, and inferences may be made about the differences in locations between respondents and items (including levels within items). A respondent's location on the scale is often interpreted as a "proficiency" or "ability", a term from academic testing, since in that case it represents their likelihood of responding correctly to more items. The zero value on this scale represents the average location of all respondents. The "location" of an item is a real number that indicates the difficulty to rate that item correctly. Higher values correspond to higher difficulty. If the person location is equal to the item location, then that person has an ability equal to that item's difficulty. If a person location is higher it has a greater ability than that item's difficulty. If the person location is lower, it has a lesser ability than that item's difficulty. The item-characteristic curve (ICC) for the 1pl is shown in Figure 2.

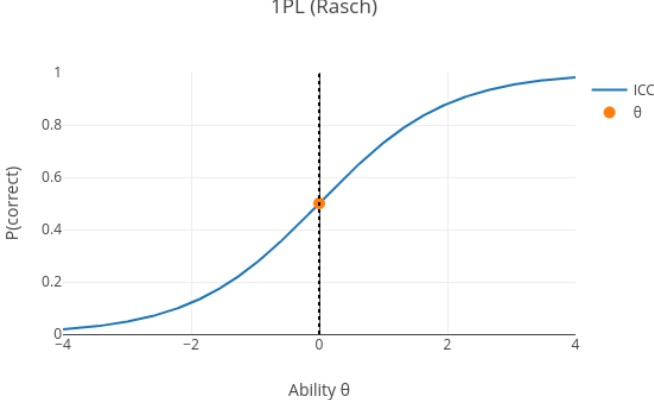

Figure 2: The Item Characteristic Curve (ICC) for the 1PL Rasch Model. It shows the Ability, $\theta$, plotted against the probabilistic scale, estimating the likelihood of getting an item correct.

## 3 RELATED WORK

We are not the first to notice the utility that IRT could have in the ML space. An early, notable work, from 2023, Narins et al. (2023), used IRT in the NLP space for validating an image-caption rating dataset. Another work took place at a workshop at the 2024 European Chapter of the Association for Computational Linguistics (EACL) conference. At the conference, Lalor, et. al. covered their Python library, `py-irt`, and applying IRT to the NLP space in multiple ways, including classification, data validation, a way to find annotation error, and using $\theta$ as an alternative to accuracy for ranking Lalor et al. (2024a;b); Lalor & Rodriguez (2023). Another work, Zhou et al. (2025), also uses $\theta$ for ranking, and compares it with popular benchmarks, finding that in some cases ranking by $\theta$ changes the ordering on a leaderboard. Both, Lalor et al. (2024a) and Zhou et al. (2025) discuss how benchmarks may be misleading and how IRT can provide more information than traditional metrics alone.

In the last couple years, there have been several new and intriguing research developments applying Psychometric practices and IRT to NLP problems for evaluation, calibration, validation, and alignment Liu et al. (2025); Sharpnack et al. (2024); Song et al. (2025); Veeramani et al. (2024); Jain et al. (2025); Narins et al. (2023). We believe this is a result of the plethora of models we have to choose from, the need to discern between them, and because traditional evaluation metrics and benchmarks that do not include item information could be misleading Zhou et al. (2025); Schilling-Wilhelmi et al. (2025); Sarkar et al. (2025); Ding et al. (2024). Many researchers are discovering and exploring the use of IRT when using LLM-as-a-Respondent and LLM-as-a-Judge Scarlatos et al. (2025); Gurdil et al. (2024). And some are using Pyschometrics and IRT to explore the psychological aspect of AI Federiakin (2025); Vogelsmeier et al. (2025); Chollet (2019); Zhuang et al.; Ye et al. (2025).

## 4 METHODS

In the following sections, we present two case studies utilizing VLMs-as-a-Respondent, comparing and analyzing against a ground-truth of humans. Using a 1PL Rasch model with Wright Map not only allows us to find the best VLM for a ranking task, but also provides an interpretable analysis of our item difficulties, person abilities, construct map, and scale. Combining all of these allows us to ensure the internal structural validity of the latent trait we are modeling. In our work, all Rasch model estimations were generated by using ACER Conquest 5.0 Adams et al. (2012).

Performing a complete IRT and Wright Map analysis involves collecting data, running statistical inference to find $\theta$ and $\delta$, checking item fit statistics, checking internal reliability and structure of the scale, generating a Wright Map with waypoints and plotting individual locations of respondents of interest, then interpreting the results in the context of the latent ability that is being assessed Wilson & De Boeck (2004); Wilson (2023).

## 5 VISION LANGUAGE MODELS

We used 10 popular VLMs from 6 vendors. We wanted to use a variety of models based on parameter scale, architecture, and training strategies, but we also had to work within budget and hardware availability. The models used in the study are below. We'll describe each model briefly. Note that not all models were able to perform in both case studies [1].

*SmolVLM-Instruct* Research (2024) This model was developed by HuggingFace to be an open, compact model. It has 2.25 billion parameters, and was optimized for instruction-following and fast inference. It was released on November 18, 2024.

*Phi-3.5-Vision-Instruct* Microsoft (2024) This model was developed by Microsoft to be an open, mid-sized model. It has 4.2 billion parameters, and was trained on synthetic and filtered public data. It was optimized for instruction-following with images. It was released on August 20, 2024.

---

[1] SmolVLM-Instruct was unable to process text-only input, and LLAMA3.2-11B-Vision-Instruct was unable to handle multiple images, both in the visual reading comprehension study.

*Qwen2-VL-7B-Instruct* Team (2024) This model was developed by Alibaba Cloud to be an open, large model with 7 billion parameters. It was optimized for instruction-following with images. It was released on October 3, 2024.

*Gemini-1.5-flash* AI (2024) This model was developed by Google. It is proprietary but we know it is based on a sparse mixture-of-experts (MoE) transformer architecture. The parameter count has not been disclosed. It was released on May 14, 2024.

*Llama3.2-11B-Vision-Instruct* Meta Platforms (2024) This model was developed by Meta to be an open, large model with 11 billion parameters. It was optimized for instruction-following with images. It was released on September 25, 2024.

*GPT-4o-2024-08-06, GPT-4o-2024-05-13, GPT-4o-2024-11-20, GPT-4o-mini, GPT-4-turbo-2024-04-09* OpenAI (2025) These models represent different iterations and parameter scales of the GPT-4 family. The "o" (omni) series allows for processing image embeddings alongside textual prompts, while the "mini" variant serves as a parameter-efficient version. The "turbo" variant has optimizations in speed and context handling compared to earlier GPT-4 models. GPT-4o-mini was released on July 18, 2024.

# 6   CASE STUDY 1: IMAGE CAPTION RATING

Automatic Generation of Image Captions and Video Descriptions is essential for visual retrieval, recommendation engines, and accessibility for Blind and Low Vision (BLV) individuals Karpathy & Fei-Fei (2015); Datta et al. (2008); Crudden & McBroom (1999); Kirchner & Smith (2005); McDonnall & Sui (2019); McMillen & Alter (2017); Shaw et al. (2007). Assessing the quality of the image captions and video descriptions is an increasingly important area of research as more generative solutions come online. Given the limitations of existing metrics, direct human evaluation remains the gold standard, but it is costly, subjective, and time-consuming to run data collection with humans-in-the-loop. The Validated Image Caption Rating (VICR) study addressed some of these challenges by introducing a high quality dataset and a well-defined rating scale, calibrated with IRT, which reflects human assessments of the quality of image-caption pairs, specifically for the BLV community Narins et al. (2023); Scott et al. (2023). We extend the VICR work with the addition of VLM raters to do the same task as the human raters. We then analyze the VLMs with IRT and a Wright Map with waypoints in conjunction with the original human raters.

The VICR dataset uses a detailed and robust rating scale, which was designed to capture the extent of four essential aspects of image captions: accuracy, completeness, local context, and global context and inferential information. The authors of the VICR scale claim it is based on sociocognitive theories Blum et al. (2020); Kintsch (1988); Smith & Hancox (2001). In their study, they selected 25 image-caption pairs from their dataset, originally used as tutorial examples in a gamified, rating task known as the "Rating Game." The 25 items were comprised of 5 distinct images, each paired with 5 distinct captions that reflected the 5 levels of coherence on the scale, ranging from level 1 (lowest coherence) to level 5 (highest coherence). All captions were carefully selected by multiple domain experts to represent the full spectrum of quality.

The original Rating Game collected 132 human responses. We then prompted 10 VLMs to rate the same 25 image-captions pairs. We combined the human and VLM-generated ratings for the 1PL Rasch analsysis, and investigate not only whether VLMs can serve as valid substitutes for human raters, but also whether the 25 items are sufficiently informative for evaluating rating behavior. The prompt given to the VLMs is shown in Appendix A. Essentially, they were prompted to play the role of an expert image-caption rater for visually impaired and blind users, and requested to rate each image-caption pair using the scale provided. Also see Appendix A for for the scale.

Since prompts are highly subjective and can significantly influence model performance, we designed our prompts by following the "Hierarchical Prompt" methodology from the ShareGPT4V Chen et al. (2024) prompt template as a foundation. Each VLM was assigned a specific role, treating them as a "character" within the prompt. The prompts are structured to optimize the model's performance in assessing caption quality. First, we define the role of the model, explicitly guiding it toward the task of rating captions. Then, we provide the rating scale as a reference framework to standardize its evaluations. Finally, we constrain the model's task to solely generate ratings. We also provide the

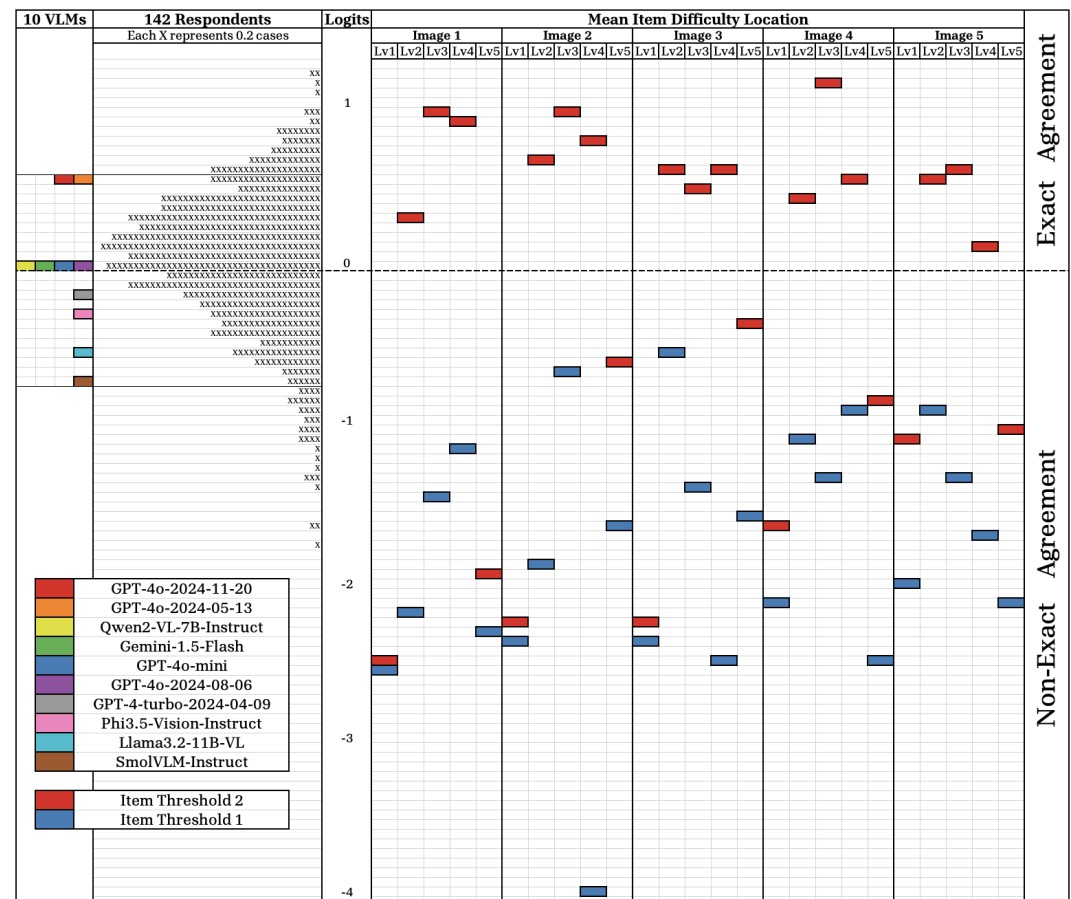

Figure 3: The Wright Map for the Image-Caption Rating task. The distribtion of $\theta$ abilities for respondents is shown under the column "142 Respondents". The VLMs are included in this group and additionally pulled-out into their own column, labeled "10 VLMs" so they can be seen in isolation. The item-map, on the right side, shows each of the 25 items (image-caption pairs) and their location. You can see "banding" of red and blue items. The waypoint separates the bands showing where respondents had Exact Agreement with the ground-truth or Non-Exact.

base64-encoded image and associated caption in each prompt. Each rating was generated independently, without the use of any memory, to avoid any carryover effects.

## 6.1 IRT AND WRIGHT MAP ANALYSIS

Figure 3 shows the Wright Map for the results we obtained from the 1PL for the image-caption rating task. In our analysis, all 25 items had fit statistics within the acceptable range so we can trust the validity. The Partial Credit Model (PCM) was used to label the responses Wright (1982) at 3 levels of agreement: level 0 is called "Distal Agreement", meaning a rating two units or more away from the correct rating. Level 1 is called "Adjacent Agreement", meaning a rating exactly one unit above or below the correct rating. Level 2 is called "Exact Agreement", meaning that the respondent gave the correct numerical rating as the value provided by the expert. Consequently, there are 2 cumulative thresholds, between these 3 levels Wilson (2023). The Wright Map empirically represents the hypothesis that the levels of agreement defined above (distal, adjacent, exact) behave consistently across items.

Respondents aligned with *Item Threshold 1* for a given item have a 50% chance of scoring in the lowest category (e.g., level 0 representing Distal Agreement) and a 50% chance of scoring into any of the upper levels (Adjacent Agreement and Exact Agreement) for that item. Respondents aligned with *Item Threshold 2* for an item have a 50% chance of scoring level 0 and 1, and 50% chance of scoring into level 2 (Exact Agreement). Respondents who are above a threshold have more than 50%

| Vision Language Models | Ability ($\theta$) | Percentile |
|---|---|---|
| GPT-4o-2024-11-20 | **0.52807** | **89%** |
| GPT-4o-2024-05-13 | **0.52807** | **89%** |
| Qwen2-VL-7B-Instruct | 0.01775 | 54% |
| GPT-4o-mini | 0.01775 | 54% |
| GPT-4o-2024-08-06 | 0.01775 | 54% |
| Gemini-1.5-flash | -0.09786 | 41% |
| GPT-4-turbo-2024-04-09 | -0.21015 | 33% |
| Phi-3.5-Vision-Instruct | -0.31962 | 24% |
| Llama3.2-11B-Vision-Instruct | -0.53158 | 16% |
| SmolVLM-Instruct | -0.73605 | 12% |

Table 2: Evaluation of each VLM on the image caption rating task. The columns labeled "Ability" and "Percentile", indicating the logit location for each VLM (its ability) and the percentage of the human raters that the VLM matched or exceeded in ability.

chance of being correct on that item at the level corresponding to that threshold; whereas respondents who are below that threshold have less than a 50% chance. Overall, respondents that are higher on the scale have more ability at rating image-caption pairs. Items and their associated threshold, that are high on the scale are considered more difficult for respondents to achieve "Exact Agreement" (e.g. level 2). The inverse holds true for respondents and item thresholds that are comparatively lower.

The column labeled "10 VLMs" shows the location of the ten VLM respondents on the scale. The column labeled "142 respondents" shows the location of all 142 respondents, which includes the 10 VLMs and the 132 human respondents. As a reminder, a higher location for a respondent indicates higher proficiency with respect to latent ability of rating image-caption items (image-caption pairs). The 25 columns on the right, grouped under the heading "Mean Item Difficulty Locations" correspond to the 25 items that we used in constructing this map. Each column corresponds to one item, i.e., one image-caption pair. Each column indicates with a red rectangle the location of Item Threshold 2 for that item, and with a blue rectangle the location of Item Threshold 1 for that item.

One of the observations we see on the Wright Map is that Items representing the middle categories (i.e. items with VICR ratings of 2, 3, or 4) are harder to get exact agreement than the extremes (e.g., items with VICR ratings of 1 or 5). This shows that the appropriate score for mid-quality captions is harder for individuals to discern than for very high or very low quality captions.

GPT-4o-2024-11-20 and GPT-4o-2024-05-13 had the highest $\theta$ (ability) out of all VLMs; these top two VLMs also performed better than or equal to 89% of the human respondents. There are 4 other VLMs Qwen2-VL-7B-Instruct, Gemini-1.5-Flash, GPT-4o-mini, and GPT-4o-2024-08-06 that scored close to 0, suggesting that those VLMs are as good as the average human respondent. Moreover, GPT-4-turbo-2024-04-09, Phi-3.5-Vision-Instruct, Llama3.2-11B-Vision-Instruct, and SmolVLM-Instruct performed below the average respondent suggesting they are not as good at the image-caption rating task, relative to the other respondents.

# 7 CASE STUDY 2: VISUAL READING COMPREHENSION

In Blum (2019), the author set out to investigate the impact of visual reading material on comprehension, more specifically, degree of information integration Blum et al. (2020). Notions of accessibility bring to question the perceived deficits in narrative comprehension for autistic people. This deficit has been positioned as having a cognitive processing disposition towards local inference (taken directly out of the material), rather than global inference (requiring greater integration of world knowledge). Instead of a unitary deficit in the individual, reduced performance on inferential narrative comprehension tasks was hypothesized to be an issue of modality; that is, does the presence of visual information impact the results? In the study, the impact of modality on inferential reasoning was compared between autistic and non-autistic adolescents.

The original study included $N = 130$ human respondents (18 autistic, and 112 non-autistic). The author constructed narratives that focused on moral and ethical issues involved in human relationships (e.g., friendship, stealing, lying). These narratives were presented in two formats: unimodal text-only and multimodal image plus text. They are both word-for-word the same, except one is a

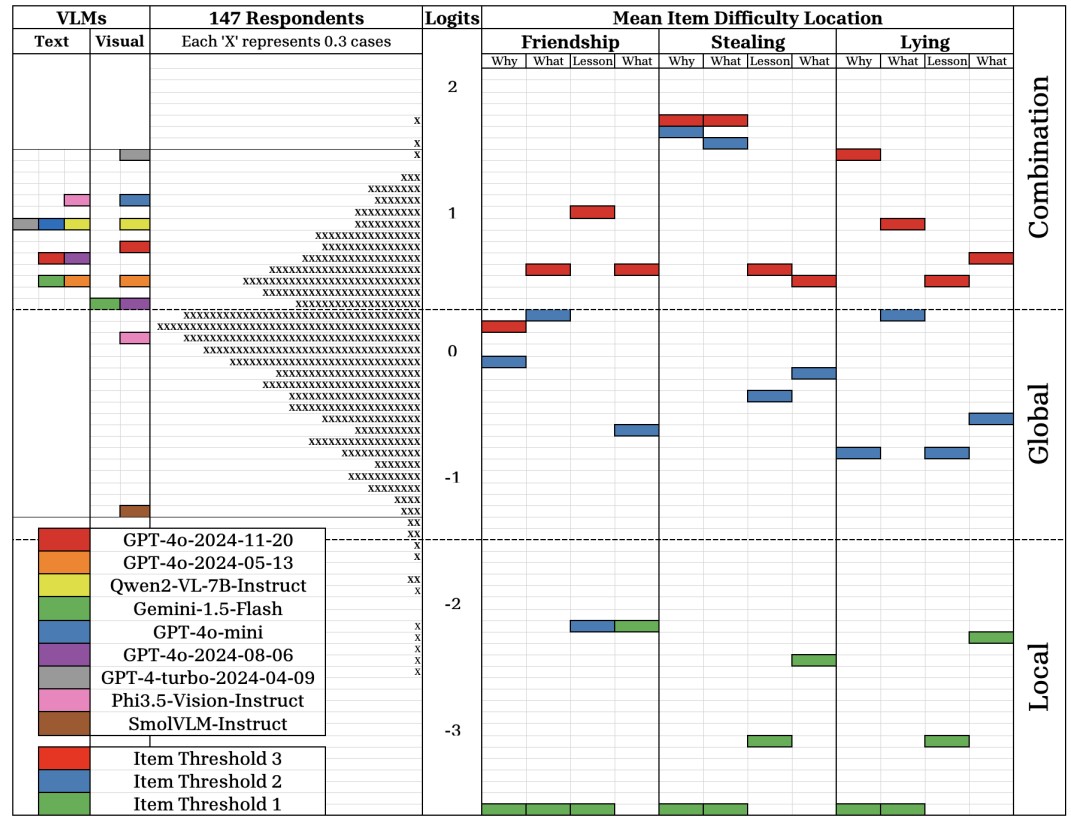

Figure 4: The Wright Map for the visual reading comprehension task. The column "147 Respondents" shows the distribution of the $\theta$ abilities at information integration for all respondents, humans and VLMs, and in both formats. The VLMs are pulled-out into their own column and format so they can be seen in isolation. The item-map on the right side shows the three narratives, and their corresponding item locations. There is fairly clear "banding" showing the validity of the latent ability's internal structure.

text format and the other has the same words spread across panels within narration boxes, speech bubbles, and thought bubbles within the images.

We prompted the VLMs with the prompt in Appendix B to engage in the same task as the human respondents. For each narrative story, the prompt had the VLMs adopt the role of a grade-school student, had them "read" thenarrative and then answer the subsequent chain of questions, presented in text-only and visual plus text formats. The three narratives were stories about a boy's friendship with a cat, a boy stealing someone's wallet for a bike, and a friend lying to protect their friend's cheating.

The initial question asked of the respondents was situated in the narrative, and the subsequent questions delved into their reasoning. The questions involved were: 1. A motivational inference question about why a character engaged in an intentional action. 2. A meta-cognitive question, asking *What made them think of the answer?* 3. An evaluative inference question, asking *What lesson could someone learn from this story?* 4. Another meta-cognitive inference question, asking *What made them think of the answer?* Human and VLM responses were encoded as Local Inference, Global Inference, or a Combination of them both.

In the study, each human group was randomly assigned to a sub-group (groups A, B, C, and D). Groups A and B consisted of randomly assigned autistic individuals. Group A read the narratives in visual format. Group B read the narratives in text-only format. Groups C and D consisted of randomly assigned non-autistic individuals. Groups C read the narratives in visual format and group D read it in text-only. The VLMs were assigned both formats.

| Vision Language Models | Ability ($\theta$) | Percentile |
|---|---|---|
| GPT-4-Turbo-2024-04-09 (comic) | **1.55903** | **100%** |
| GPT-4o-mini (comic) | 1.18214 | 97% |
| Phi-3.5-Vision-Instruct (text) | 1.18214 | 97% |
| GPT-4-turbo-2024-04-09 (text) | 1.00795 | 93% |
| GPT-4o-mini (text) | 1.00795 | 93% |
| Qwen2-VL-7B-Instruct (comic) | 1.00795 | 93% |
| Qwen2-VL-7B-Instruct (text) | 1.00795 | 93% |
| GPT-4o-2024-11-20 (comic) | 0.84269 | 88% |
| GPT-4o-2024-08-06 (text) | 0.68559 | 82% |
| GPT-4o-2024-11-20 (text) | 0.68559 | 82% |
| GPT-4o-2024-05-13 (comic) | 0.53525 | 78% |
| GPT-4o-2024-05-13 (text) | 0.53525 | 78% |
| Gemini1.5_flash (text) | 0.53525 | 78% |
| GPT-4o-2024-08-06 (comic) | 0.38988 | 67% |
| Gemini1.5_flash (comic) | 0.38988 | 67% |
| Phi-3.5-Vision-Instruct (comic) | 0.10637 | 48% |
| SmolVLM-Instruct (comic) | -1.25288 | 5% |

Table 3: Evaluation of each VLM on the visual reading comprehension task. The columns labeled "Ability" and "Percentile" indicate the logit location for each model (its ability) and the percentage of human raters that the model matched or exceeded in ability.

### 7.1 IRT AND WRIGHT MAP ANALYSIS

From analyzing the output of the statistical inference, we find that the fit statistics were within acceptable range, and we could trust the validity of the Wright Map for this latent ability. The results of the IRT and Wright Map for the Visual Reading Comprehension study are shown in Figure 4. We see on the map that most VLMs are above average, compared to human respondents, and that most VLMs are in the same region of interest. Since they are all above the 0 logit, the results suggest that all of the VLMs are good at the comprehension task, regardless of modality, with the exception of SmolVLM and Phi3.5.

Although most of the VLMs are in a similar region, regardless of format, three of them stand-out as anomalous: GPT-4-Turbo-2024-04-09, Phi3.5-Vision-Instruct, and SmolVLM-Instruct. GPT-4-Turbo-2024-04-09 has the highest latent ability with the visual format part of the task, even exceeding the human respondents, but performed marginally lower on the text format part – but not so low as to be in a different level of information integration. Whereas, Phi-3.5-Vision-Instruct measured as the first highest in text-only format, but performed significantly lower in the visual format, placing at a different region. SmolVLM-Instruct performed far below average, compared to human respondents and other VLMs. It would not be a good candidate, relatively speaking, at the visual reading comprehension task.

## 8 DISCUSSION AND CONCLUSIONS

In this paper we showcase the evaluative and interpretative power of Item Response Theory for the ML and NLP communities by using 10 Vision Language Models on two case studies: image caption rating and visual reading comprehension. Traditional accuracy-focused metrics show the general overview of model performance but lack detail provided by considering item difficulty when assessing respondent ability, such as $\theta$, $\delta$, fit statistics, and the validity of internal structure of the latent ability.

By employing the Rasch model and Wright Map analyses, we demonstrated a comprehensive and informative evaluation strategy that extends beyond aggregate performance, providing valuable insights into the latent abilities of VLMs compared to human respondents. Through our IRT investigation, we were not only able to uncover how well each of the 10 VLMS performs on each task but also how hard the items are to the respondents. Incorporating the principles and practices of IRT in the NLP problem space for the ML Community could offer significant methodological advantages for validating both items and respondents. We hope that our work may prove a valuable bridge between the measurement and machine learning communities.

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

## A    APPENDIX A: CASE STUDY 1 PROMPT

The "<Image>" and "<Caption>" tags are replaced with the base64 encoded image, and the caption.

---

You are an expert at determining how good a description is for an image. You are able to rate the given description and image on a scale of 1 to 5, corresponding to how well the description fits the image. Using the following rubric of rating levels:

1. Objects are incorrectly identified. The caption gives the wrong idea about what is happening in the image.

2. Objects are partially correctly identified with some errors, but the caption is accurate enough to give an idea of what is happening in the image. The caption identifies most of the objects but might not identify everything. There is no interpretation of what anything means.

3. Relevant objects are correctly identified. The caption describes what is seen but not where objects are in space. There is no description of the overall setting and no interpretation of an event.

4. Objects and/or a general scene and/or an action are correctly identified but not every element is completely identified. The caption describes what is seen and where things are in space. There is no interpretation of an event.

5. Objects, a general scene, and actions are correctly identified if present in the image. The caption describes what is seen and where things are in space. Interpretation of overall setting and/or event is included.

Please rate the following description on how well they describe the given image. Just the number (ratings from 1 to 5).

<Image>
<Caption>

---

## B    APPENDIX B: CASE STUDY 2 PROMPT CHAIN

The prompt used for Case Study 2 was a scripted, multi-prompt. First we would prompt with the initial context, ask a question, wait for response, and then asked another question. Here is the chain of questions.

---

You are a middle school or high school student. Please answer the following questions based on the given narrative in 1 sentence.

<Comic Panels>

**Question:** Why did the red headed boy keep the cat?
**Assistant:** <Response 1>

---

↓

---

**Question:** What made you think of that answer?
**Assistant:** <Response 2>

---

↓

**Question:** What lesson could someone learn from this story?
**Assistant:** <Response 3>

↓

**Question:** What made you think of that answer?
**Assistant:** <Response 4>

