# OpenReview forum: "Evaluation of Vision Language Models with Item Response Theory"
_ICLR.cc/2026/Conference — ICLR 2026 Conference Withdrawn Submission_

### Official Review · Reviewer_QcqE · 2025-10-26

**Soundness:** 3
**Presentation:** 2
**Contribution:** 4
**Rating:** 6
**Confidence:** 2

**Summary:**

This paper proposes a more fine-grained evaluation approach for vision–language models by weighting instances based on their difficulty. Using the Rasch model and Wright Map analyses, the authors introduce an evaluation framework that extends beyond traditional aggregate performance metrics (accuracy,...), which have been proven to be flawed.

**Strengths:**

- The paper presents an **innovative and original perspective** on VLM evaluation by explicitly modeling the relationship between item difficulty and model ability, offering a more adaptive way to assess performance.

- It provides two clear and instructive examples of how Wright Maps can be used for evaluating VLMs, serving as a strong demonstration of the potential of IRT-based analysis in this context.

- The study is supported by a robust empirical setup, including a diverse set of VLMs of varying scales and capabilities.

**Weaknesses:**

- While the use cases are interesting, it remains unclear how the proposed methodology can **generalize to other VLM tasks**. Despite the broader claims in the introduction. The authors should better articulate the **practical requirements for applying this method**, such as the need for ordinal human ratings, a sufficient number of annotators, and other dataset characteristics.

- It would be particularly helpful to include a categorization of VLM task types according to their suitability for this kind of evaluation (e.g., easy to apply, moderately adaptable, currently infeasible due to lack of ground truth or rating structure).

- The paper’s impact would be greatly enhanced if the authors provided a tool or codebase to enable large-scale evaluation across multiple tasks.

- As it stands, the conclusions are primarily drawn from a single example setting (image caption rating and visual reading comprehension). The generalizability of the findings would be stronger if the paper included additional examples or replications of the analysis.

**Questions:**

Q1: Was the difficulty of a task estimated solely from human respondents (e.g., based on their agreement), or were model responses also included? Including models might bias the estimation of item difficulty, so clarification on this design choice would be important.

Q2: Has a similar IRT-based methodology been previously applied to LLM evaluation tasks? A discussion of related applications in the language domain could help contextualize the contribution.

Q3: What kinds of tasks are best suited for this type of evaluation, and what are the practical requirements (data type, annotation structure, etc.) to make it feasible?

Q4: Could the authors provide more insight into what makes a task difficult or what differentiates more capable VLMs in their framework?
For instance, any intuition on why GPT-4o-Turbo performs particularly well on visual reading comprehension but poorly on image caption rating would be valuable.

---

### Official Review · Reviewer_gKV4 · 2025-10-26

**Soundness:** 1
**Presentation:** 2
**Contribution:** 2
**Rating:** 2
**Confidence:** 3

**Summary:**

The paper proposes an IRT-based evaluation for VLMs that jointly models ability and item difficulty on a shared logit scale, visualized via Wright Maps and validated with MNSQ fit. Two case studies reveal which models are strong even on hard items, insights that accuracy-only metrics miss.

**Strengths:**

1. This paper proposes a difficulty-aware, interpretable metric.

2. Two case studies are interesting and thus necessitating a difficulty aware evaluation metric.

**Weaknesses:**

1. Lack of scaled evaluation. The central claims rest on two small case studies rather than dataset-scale experiments, and no baseline comparisons are reported (e.g., difficulty-bucket accuracy)

2. Unverified modeling assumptions. Rasch 1PL/PCM presumes unidimensionality and local independence, assumptions that are questionable for multi-skill VLM tasks.

3. Limited external validity. The chosen case studies (caption rating, short reading) may not generalize to grounding, chain-of-thought VQA, or instruction-following, where skills and item structures differ substantially.

4. Ablations on the IRT choice are missing. It’s unclear whether results depend on 1PL vs. 2PL/PCM choices.

5. Using VLMs and a subset of humans as “raters” risks bias.

**Questions:**

Please refer to weakness.

---

### Official Review · Reviewer_pGyV · 2025-10-27

**Soundness:** 1
**Presentation:** 2
**Contribution:** 1
**Rating:** 2
**Confidence:** 3

**Summary:**

This paper uses Item Response Theory (IRT) and Partial Credit Model (PCM) to evaluate VLMs alongside human raters on two tasks: image caption rating and visual reading comprehension. Results show that using partial credit model could better reflect VLMs' abilities compared to accuracy based methods.

**Strengths:**

1. The IRT framing makes model–item comparisons interpretable, and Wright Maps help read where models sit against humans and item thresholds. This supports qualitative insights like the difficulty of mid‑quality captions.
2. The authors benchmark 10 VLMs and show more insights than accuracy based metrics.

**Weaknesses:**

1. This paper is more like a report paper, however, the results may be incomplete: no standard errors or separation reliability are shown. Without these, ability differences and group comparisons are hard to identify. Additionally, the results can be dependent on the prompting.
2. The used item pools are too small, and only 2 VLM tasks are involved, making the results and analysis insufficient to draw valuable conclusions.
3. The evaluated models may be somewhat outdated. Better to include more recent models such as Gemini2.5 and GPT-5. Also it is necessary to test on more benchmarks and datasets.

**Questions:**

Please see the weakness.

---

### Official Review · Reviewer_aWuw · 2025-10-29

**Soundness:** 2
**Presentation:** 2
**Contribution:** 2
**Rating:** 4
**Confidence:** 2

**Summary:**

This paper argues that accuracy-oriented benchmarks provide an incomplete picture of model performance because they fail to incorporate the difficulty of items. To address this, the authors propose using Item Response Theory (IRT) to evaluate the output of Vision Language Models (VLMs). IRT allows for the simultaneous measurement of the VLM’s latent "ability" and the test item’s "difficulty" on the same scale, a method validated across two key tasks: image caption rating and visual reading comprehension.

**Strengths:**

1. This paper reveals that accuracy-oriented benchmarks may provide an incomplete picture of model performance because they fail to incorporate the difficulty of items.
2. This paper proposes to use Item Response Theory (IRT) to evaluate the output of VLMs, which enhances interpretability.

**Weaknesses:**

1. The paper fails to provide a crucial comparison between the model rankings derived from IRT ability estimates and those from traditional accuracy-oriented metrics. If the correlation is high, the practical value of IRT for ranking models is limited.
2. The methodology requires highly specialized and manual prompt engineering (e.g., multi-step prompt chains), which significantly hinders the scalability and convenience for general ML workflows.
3. The analysis is limited to only two specific case studies, which restricts the generalizability of the IRT framework as a general evaluation tool.

**Questions:**

See weaknesses

---

### Note · Authors · 2025-11-20

**Comment:**

Dear Reviewers,
Thank you so much for your time reviewing our submission. We have taken your comments and suggestions into consideration and will address them in future work. For now, we withdraw this submission.
Sincerely,
First Author and Team

**Withdrawal Confirmation:**

I have read and agree with the venue's withdrawal policy on behalf of myself and my co-authors.